# Clinical Profile of SARS-CoV-2 Infection: Mechanisms of the Cellular Immune Response and Immunogenetic Markers in Patients from Brazil

**DOI:** 10.3390/v15071609

**Published:** 2023-07-23

**Authors:** Vanessa Pacheco, Rosane Cuber Guimarães, Danielly Corrêa-Moreira, Carlos Eduardo Magalhães, Douglas Figueiredo, Patricia Guttmann, Gisela Freitas Trindade, Juliana Fernandes Amorim da Silva, Ana Paula Dinis Ano Bom, Maria de Lourdes Maia, Juliana Gil Melgaço, Tamiris Azamor da Costa Barros, Andrea Marques Vieira da Silva, Manoel Marques Evangelista Oliveira

**Affiliations:** 1Quality Assurance Department, Bio-Manguinhos–Fiocruz, Rio de Janeiro 21040-900, Brazil; vanessa.pacheco@bio.fiocruz.br; 2Vice Director of Quality, Bio-Manguinhos–Fiocruz, Rio de Janeiro 21040-900, Brazil; rosane@bio.fiocruz.br; 3Laboratory of Taxonomy, Biochemistry and Bioprospecting of Fungi, Oswaldo Cruz Institute–Fiocruz, Rio de Janeiro 21041-250, Brazil; dcorrea@ioc.fiocruz.br; 4UERJ-Universitary Hospital Pedro Ernesto, Outpatient Vascular Surgery, Rio de Janeiro 20950-003, Brazil; cevirgini@gmail.com (C.E.M.); dpvasc@yahoo.com (D.F.); 5Municial Health Secretary of Rio de Janeiro, Rio de Janeiro 20211-110, Brazil; patguttmann@gmail.com; 6Laboratory of Virologic Tecnology, Bio-Manguinhos–Fiocruz, Rio de Janeiro 21040-900, Brazil; gisela.freitas@bio.fiocruz.br (G.F.T.); juliana.silva@bio.fiocruz.br (J.F.A.d.S.); 7Laboratory of Imunologic Tecnology, Bio-Manguinhos–Fiocruz, Rio de Janeiro 21040-900, Brazil; adinis@bio.fiocruz.br (A.P.D.A.B.); juliana.melgaco@bio.fiocruz.br (J.G.M.); tamiris.azamor@bio.fiocruz.br (T.A.d.C.B.); amarques@bio.fiocruz.br (A.M.V.d.S.); 8Clinic Assessory, Bio-Manguinhos–Fiocruz, Rio de Janeiro 21040-900, Brazil; lourdes.maia@bio.fiocruz.br (M.d.L.M.); paulo.takey@bio.fiocruz.br (Collaborative Group)

**Keywords:** SARS-CoV-2, COVID-19, immune response, cytokine storm, acute infection

## Abstract

Objectives: The aim of this study is to evaluate some mechanisms of the immune response of people infected with SARS-CoV-2 in both acute infection and early and late convalescence phases. Methods: This is a cohort study of 70 cases of COVID-19, confirmed by RT-PCR, followed up to 60 days. Plasma Samples and clinical data were. Viral load, blood count, indicators inflammation were the parameters evaluated. Cellular immune response was evaluated by flow cytometry and Luminex immunoassays. Results: In the severe group, hypertension was the only reported comorbidity. Non severe patients have activated memory naive CD4+ T cells. Critically ill patients have central memory CD4+ T cell activation. Severe COVID-19 patients have both central memory and activated effector CD8+ T cells. Non-severe COVID-19 cases showed an increase in IL1β, IL-6, IL-10 and TNF and severely ill patients had higher levels of the cytokines IL-6, IL-10 and CXCL8. Conclusions: The present work showed that different cellular responses are observed according to the COVID-19 severity in patients from Brazil an epicenter the pandemic in South America. Also, we notice that some cytokines can be used as predictive markers for the disease outcome, possibility implementation of strategies effective by health managers.

## 1. Introduction

In last years the world has been living a pandemic that started in December 2019, with several severe respiratory distress syndrome cases of unknown cause in Wuhan Province, China [1]. Through genetic sequencing of lung lavage, the pathological agent identified was a new coronavirus type, later named SARS-CoV-2, an enveloped single-stranded RNA betacoronavirus, whose main invasion mechanism is the binding of its structural protein S (spike) with angiotensin-converting enzyme 2 (ACE-2) on cell’s surface [1,2,3,4,5]. The virus rapid spread worldwide and led World Health Organization (WHO) to declare the occurrence of this pandemic on 11 March 2020 [6].

COVID-19 understanding is evolving over time. The range of symptoms varies, from fever, dry cough, sore throat, dyspnea, fatigue, myalgia and diarrhea to severe form, with pulmonary involvement, in 20% of the patients. Headache is one of the most prevalent symptoms with a strong association between rhinosinusitis and SARS-CoV-2 infection, although the pain mechanism likely resides in a systemic reaction to the virus. Nasal symptoms have already been mentioned, and some authors speculate whether the causes of this cluster of symptoms may be due to activation of the trigeminal autonomic reflex by central or meningeal negotiation, or even direct viral damage to the central or peripheral nervous system during an infection [7].

Different variants of SARS-CoV-2 have been associated with different risks and illness’ severity [8,9,10,11]. The increased risk of death can be associated with some risk factors such as cardiovascular disease, diabetes mellitus, arterial hypertension, chronic obstructive pulmonary disease, neoplasms, chronic renal failure, obesity, smoking, male gender and advanced age [3].

Regarding the mechanisms of the immune response against this virus, it is important to mention that higher concentrations of cytokines as granulocyte-colony stimulating factor (G-CSF), interferon gamma-induced protein 10 (IP10), monocyte chemoattractant protein 1 (MCP1), macrophage inflammatory protein 1alpha (MIP1A), tumor necrosis factor alpha (TNFα) interleukin-2 (IL-2) receptor, interleukin-6 (IL-6), interleukin-8 (IL-8), interleukin-10 (IL-10), were reported in patients with COVID-19 [12,13]. These reactions characterize the cytokine storm, a disordered systemic response that leads to a hyperinflammation condition in the host and culminates in an untoward clinicopathological consequences [11]. Some authors describe that observed in infected patients an increase of MIP-1α levels, a cytokine involved in lymphocyte and monocyte endothelial attraction and migration [14]. This may explain the extreme lymphopenia with reduced CD4 and CD8 T populations, that has been shown to be a consistent prognostic factor in patients with severe forms [15,16].

Furthermore, it is important to mention the role of immune mechanisms in patients with severe disorders, such as cancer or other immunosuppressive conditions. The T-cell repertoire in their patients was skewed towards differentiated phenotypes expressing IFNγ, but even more pronounced towards IL-17 production, since SARS-CoV-2 infection induced a Th17 signature, which very likely contributes to disease severity through exacerbated inflammation. Additionally, they show elevated percentages of circulating neutrophils, which is a signature of dysfunctionality and elevated baseline inflammation [17].

Based on this, the aim of this study was to demonstrate some mechanisms involved in the immune response of a cohort of severe and non-severe COVID-19 patients, through the analysis of lymphocyte subpopulations and the profile of cytokines produced by these patients.

## 2. Materials and Methods

### 2.1. Study Design and Participants

This is a cohort study of 70 cases of COVID-19, confirmed by RT-PCR, followed up to 60 days. A convenience sample of 70 COVID-19 cases was stratified into severe (30) and non-severe (40). The non-serious group was defined with a greater number of participants due to the possibility of group migration throughout the study, depending on the disease evolution. In addition, 20 healthy participants were included for laboratory method control.

The clinical study protocol was approved by Research Ethics Committee at Pedro Ernesto University Hospital/UERJ n°. 4.160.423, of 17 July 2020 and by Research Ethics Committee of SMS/RJ, approved (number 4.322.297, of 10 June 2020). According to the Declaration of Helsinki (2008) and Resolution number 466 of National Health Council (2012), confidentiality of patient information is guaranteed.

### 2.2. Data Collection of COVID-19 Patients

Plasma Samples and clinical data were collected at four time points from the date of symptom onset: 4–6, 8–10, 15–20, and 45–60 days. Cellular immunity, viral load, blood count, indicators of liver and kidney function and inflammation were the parameters evaluated. Non-serious cases were captured among symptomatic patients, considered suspects, who contacted the Research Center, or who sought care at the Pedro Ernesto University Hospital or the Piquet Carneiro Polyclinic. Severe cases were captured among patients hospitalized at the Hospital Universitário Pedro Ernesto/UERJ and in Municipal Health Department of Rio de Janeiro (SMS-RJ) health units. The group of participants hospitalized in units of SMS-RJ had the visits carried out by the study central team and the samples collected at the place of hospitalization. The other collections were scheduled according to the symptoms onset day. Not all critically ill patients had the initial samples collected, as they were included after the collection period.

### 2.3. Inclusion and Exclusion Criteria

Inclusion and Exclusion criteria are described in Table 1. Briefly, after selecting the 70 participants, 20 healthy participants were also included, with a proportional sex and age distribution, like the participants with COVID-19 included in the study, without a diagnosis of COVID-19 (with undetectable RT-PCR and IgM and IgG non-reactive), which constituted a laboratory method control group for the cellular and immunogenetic immunity evaluation. These participants were included in the same research center and only one swab and blood collection were performed on the inclusion date.

### 2.4. Immunophenotyping

Peripheral blood mononuclear cells (PBMC) were obtained from whole blood using Histopaque^®^ and Ficoll (Sigma-Aldrich, Saint Louis, MO, USA) different density gradients. These cells were cryopreserved, and then thawed at the time of each assay. Then, was used a concentration of 2 × 10^5^ viable cells/mL, and submitted to immunophenotyping assay with surface antibodies for 20 min at 2–8 °C. After, the cells were washed with phosphate buffer plus fetal bovine serum (FBS) (PBS pH 7.4 at 2% FBS), and centrifuged at 400× *g* for 5 min. After centrifugation, cells were fixed with 1% paraformaldehyde solution and subsequently acquired in a flow cytometer (LSR Fortessa^TM^, BD Biosciences, Franklin Lakes, NJ, USA). The analysis was performed using Flow Jo software v10.6 (BD Biosciences).

The anti-human antibodies used in the immunophenotyping assay were: panel I (activation)-CD3-FITC, CD4-APCH7, CD8-BV605, CD38-PECy7, OX40-BV711 and panel II (memory)–CD3-APC-Cy7, CD4-BV421, CD8-BV605, CD45RA-APC, CCR7-BV510 (BD Biosciences).

### 2.5. Cytokine Detection

#### 2.5.1. Immunospot Assay

The frequency of interferon gamma (IFN-γ) and IL-10 secreting cells in patients’ PBMCs were analyzed using the FluoroSpotplus kit human assay (Mabtech, Stockholm, Sweden) as recommended by the manufacturer. Cell suspensions were plated (2 × 10^5^ cells/well) on pre-coated plates and cultured for 20 h in the presence or absence of SARS-CoV-2 Antigen Peptide NCAP-2mcg/mL (nucleocapsid peptides, JPT peptides, Berlin, Germany). As a positive control, cells were incubated with 2 µg/well of Concanavalin A (Sigma-Aldrich). After incubation and development, the “spots” of the cells secreting the said mediators were quantified using the ImmunoSpot^®^ (CTL) image analyzer. The number of “spots” generated by cells stimulated with the antigen was subtracted from the non-specific spots generated in non-stimulated cells, generating the number of specific spots for SARS-CoV-2 per million cells.

#### 2.5.2. Multiplex Micro Array

To quantify the cytokine levels in the plasma of the patients, were used the multiplex liquid microarray assay with magnetic beads-Human Magnetic Luminex Assay (R&D Systems, Minneapolis, MN, USA) which allowed to quantify inflammatory and regulatory cytokines, IL-1b, IL-6, IL-10, IL-8/CXCL8, TNF-α. The test was performed according to the manufacturer’s recommendations. The result was performed in a MAGPIX^®^ system equipped with xPONENT v3.2 and the data were analyzed in SoftMax Pro software version 5.4, applying the five-parameter regression formula to calculate the sample concentrations from the standard curves.

### 2.6. Statistical Analysis

The results were expressed as mean and its standard deviation. Clinical characteristics of participants were compared using Mann Whitney, Kruskal Wallis, ANOVA and Spearman correlation tests. Trial results with qualitative results were presented in absolute and relative frequency, and evaluated with participants’ clinical characteristics using Mann Whitney, Kruskal Wallis, ANOVA, Chi-square and Fisher’s exact tests. Statistical analyzes with cytokine detection results were performed using the GraphPad Prism 5 software, applying one-way ANOVA, two-way Kruskal-Wallis test and Dunn’s Multiple Comparison Test to compare specific production levels of the analyzed cytokines stratified by clinical and compared to control samples. The results provided quantitative data regarding the cytokines production from specific cellular response to SARS-CoV-2.

## 3. Results

### 3.1. Investigation

The first patient was included on 8 December 2020, from UERJ, and on 10 December 2020 in RJ centers. The last research participant was included on 31 March 2021 and fieldwork also ended. Briefly, UERJ included 68 participants, and RJ centers, 28 participants. All participants had detectable PCR for SARS-CoV-2 at enrollment.

Most of the patients were male (52.1%), white (68.5%), married (47.9%) and with high level education (24.7%). The median age was 49 years, ranging from 19 to 93 years. The main symptoms among participants were fatigue, cough, headache, myalgia or arthralgia, and anosmia. Among participants with severe conditions, the most common symptoms were fatigue and dyspnea, and for non-severe ones, headache, and fatigue. Regarding the presence of comorbidities in the severe group, hypertension was reported in 51.6% of the patients and this condition was also reported by 21.4% of the patients of the no-severe group (Table 1).

**Table 1 viruses-15-01609-t001:** Distribution of participants according to medical conditions.

Medical Conditions	Severe		Non-Severe		Total		
	(N = 31)		(N = 42)		(N = 73)		
	n	%	N	%	n	%	*p*-Value *
Diabetes Mellitus							0.016
Yes	11	35.5	4	9.5	15	20.5	
No	20	64.5	38	90.5	58	79.5	
Hipertension							0.015
Yes	16	51.6	9	21.4	25	34.2	
No	15	48.4	33	78.6	48	65.8	
Obesity							0.8524
Yes	5	16.1	5	11.9	10	13.7	
No	26	83.9	37	88.1	63	86.3	
Smoking (currently)							-
Yes	0	0.0	0	0.0	0	0.0	
No	30	96.8	42	100.0	72	98.6	
Unknown	1	3.2	0	0.0	1	1.4	
Ex-smoking							0.3358
Yes	7	22.6	5	11.9	12	16.4	
No	23	74.2	37	88.1	60	82.2	
Unknown	1	3.2	0	0.0	1	1.4	
Substance abuse ou misuse							-
Yes	1	3.2	0	0.0	1	1.4	
No	30	96.8	42	100.0	72	98.6	
Special Needs/Deficiency							-
Yes	0	0.0	0	0.0	0	0.0	
No	31	100.0	42	100.0	73	100.0	
Cardiovascular Disease							0.7726
Yes	2	6.5	1	2.4	3	4.1	
No	29	93.5	41	97.6	70	95.9	
Chronic Kidney Disease							-
Yes	0	0.0	0	0.0	0	0.0	
No	31	100.0	42	100.0	73	100.0	
Chronic Liver Disease							-
Yes	0	0.0	0	0.0	0	0.0	
No	31	100.0	42	100.0	73	100.0	
Chronic Lung Disease							1000
Yes	1	3.2	1	2.4	2	2.7	
No	30	96.8	41	97.6	71	97.3	
Pulmonary tuberculosis being treated							-
Yes	0	0.0	0	0.0	0	0.0	
No	31	100.0	42	100.0	73	100.0	
Psicologic condiction							0.7726
Yes	2	6.5	1	2.4	3	4.1	
No	29	93.5	41	97.6	70	95.9	
Other chronic disease							0.7183
Yes	2	6.5	5	11.9	7	9.6	
No	29	93.5	37	88.1	66	90.4	
Other condiction							-
Yes	0	0.0	0	0.0	0	0.0	
No	31	100.0	42	100.0	73	100.0	

* Fisher’s exact test.

### 3.2. Laboratory Assays

Table 2 summarizes the laboratory analysis of the patients. We highlight that hemoglobin values were lower in severe cases and in the second week of the disease (visit 2) with a median of 10.1g/dL in critically ill patients. Critically ill patients at visit 1 also had lower lymphocyte counts with a median of 1093 cells compared to 1434 of the non-severe cases.

### 3.3. Immunophenotyping

Figure 1 shows the gate strategy to set T lymphocytes and in the Figure 2 are demonstrated percentages of activated T cells (CD38^+^ OX40^+^). The evaluation of TCD4+ cells demonstrated that severe patients had lower percentages of central and effector memory cells than naïve cells, also observed in non-severe patients. In addition, it was also possible to identify terminally differentiated cells in this group of patients. Regarding the immunophenotyping of TCD8+ cells, similarly to what was observed in TCD4+, severe patients had higher percentages of central memory cells compared to control group. Effector memory cells were identified in both groups of patients, severe and non-severe, and contrary to what was observed in TCD4+ lymphocytes, only in the non-severe group it was possible to identify naïve T cells. Finally, terminally differentiated lymphocytes were observed in the severe and non-severe groups, the former being like the control group (Figure 2).

### 3.4. Cytokine Detection

Cytokine detection: Comparing cytokine levels quantification in laboratory controls and patients in the 4/6-day collection after study admission, it was seen that non-severe COVID-19 cases showed an increase in IL1β, IL-6, IL-10 and TNF (Figure 3A–D). Severely ill patients had higher levels of the cytokines IL-6, IL-10 and CXCL8 in the first days of SARS-CoV-2 infection (Figure 2B,C,E) and, in contrast, lower levels of IL-1β and TNF (Figure 3A,D) than controls and non-severe patients. At the later time of collection, 45/60 days after admission, non-critical patients still had increased levels of IL-1β and TNF (Figure 3A,D).

According to severity, it was seen that COVID-19 severe cases have lower production of IL1β at all times analyzed (Figure 3A) and of TNF at times 4/6 and 45/60 days (Figure 3D). Concomitantly, in severe COVID-19, an increase in the initial production of IL6 and IL10 (4/6 days) was seen (Figure 3B,C) and maintenance of high levels of CXCL8 at times 4/6 and 15/20 days (Figure 3E).

After stimulation with nucleocapsid peptides (NCAP–2mcg/mL, JPT peptides), it was possible to detect IFNγ and IL10-secreting cells by the FluoroSpot technique both in the severe group and in the non-severe group (Figure 4).

FluoroSpot data showed that 22.8% of the samples had more than 10 detectable spots after stimulation with SARS-CoV-2 peptides. Therefore, the following were selected for immunophenotyping evaluation: (a) samples from 15 participants in the non-severe group who presented more than 5 spots; (b) samples of 5 serious participants who presented 2 to 3 spots; (c) samples from 5 healthy participants/controls. A difference in interferon-gamma secretion can be observed between severe and non-severe, which can be explained by the high cellular activation observed in the periphery found in the cytometry data, which can lead to cellular exhaustion and compromise the quality of interferon production.

## 4. Discussion

In this study, we set out to draw a clinical and a panel of the some mechanisms of the immune response of people infected with SARS-CoV-2 in the phases of acute infection and early and late convalescence. In this sense, we noted that the most common symptoms for severe and non-severe cases (fatigue, dyspnea, muscle pain, among others), besides comorbidities, including cardiovascular disease, diabetes, chronic respiratory disease, hypertension, frequently present in severe cases with COVID-19, were also observed by some authors that described the same profile in their patients [12,13,17]. The authors associate the COVID-19 pathogenesis with the host immune responses against the SARS-CoV-2.

In a systematic review conducted by Melo et al. [18], several cytokine storm biomarkers were described. The authors point out, among other aspects, high levels of interleukin-6, and hyperferritinemia, as well as the C-Reactive Protein, and D-dimer as important biomarkers of cytokine storm syndrome.

D-dimer is a biological marker present in blood when there is degradation of fibrin, a protein involved in clot formation. Thus, a greater amount of circulating D-dimer is associated with changes in clotting process and mainly related to an increased risk of deep vein thrombosis (DVT) and/or pulmonary thromboembolism [19]. Some authors described the increase of D-dimer was considered a infection indicator and suggest greater severity of COVID-19, since a large amount of immune response of these patients [20,21], it is important to say that were observed a decrease in the percentage of natural killer cells as well as lymphocytes cytokines is released (Cytokine storm). The mechanisms that lead to lymphopenia in COVID-19 are still not fully understood, however, the cytokine storm and, consequently, lymphocytes recruitment to inflammatory sites, apoptosis, pyroptosis and exhaustion are some hypothesis [2,19].

We also highlight hyperferritinemia observed in critically ill patients in our study, compared to non-severely ill patients. In the acute phase of the disease, which corresponds to the first collections, ferritin levels in critically ill patients were about 10 times higher than in non-severe patients. Considering ferritin as a mediator of immune dysregulation, through direct immunosuppressive and pro-inflammatory effects, this is an important predictor of cytokine storm. These data corroborate what was described by Vargas-Vargas et al. [22], in a review of clinical cases, in which elevated ferritin levels associated with diabetes and more severe outcomes of COVID-19 were observed.

Elevated serum concentrations of IL-6 and other inflammatory cytokines are hallmarks of cytokine storm and correlate with poor clinical outcomes [18]. We can cite, as an example, the high levels of C-reactive protein, a protein whose expression is driven by IL-6, as also a biomarker of severe clinical manifestations of COVID-19. Corroborating that was observed in our work, in which CRP (C-reactive protein) levels was higher in critically ill patients, and in the first visits (1 and 2). This observation is as expected, since this protein is synthesized by the liver in times of stress, especially in acute phase, such as when there is a relevant infection in progress and its function is to help the immune system, through anti-inflammatory activity [23].

Regarding some mechanisms involved in the cellular immune response in critically ill individuals in acute phase of the disease, the appearance of memory lymphocytes and antiviral cytokines after 15–20 days of viral clearance at the time of discharge of hospitalized patients [10,24]. In this study, the emergence of memory T lymphocytes was also observed in the recovery period, as well as IFNg and IL10 production. The group evaluated in the present study had a small sample size, not allowing extrapolations of results to a population scale, requiring evaluations in groups that are more representative of general population.

Cytokines play a fundamental role in COVID-19 since the severity of the disease has been associated with an exuberant production of proinflammatory cytokines, such as IL-1, IL-2, IL-6, IL-10, IL-12, IFN-γ, TNF-α, and, consequently, an excessive activation of the immune system, which may cause tissue injury, mainly on the lungs [25]. So, in the present study, were observed high levels of cytokynes, as interleukin (IL)-6, IP-10 (CXCL10), and TNFα, and proteins as C-reactive protein, ferritin, in the severe cases when compared with non-severe ones, in accordance with the description by Cao and Li [26].

According to some studies, high IL-6 levels are a signature of intense inflammatory profile in COVID-19 infections, and also a biomaker strongly related to severe siymptoms progression [10,18,27]. In this study, the circulating cytokines quantification showed that patients with COVID-19 have increased levels of IL-6 and IL-10 in the collections 4–6 days regardless of severity [9,10]. Comparing the initial cytokine levels (4–6 days) of patients according to the severity, it was seen that the non-severe were characterized by higher levels of IL1β and TNF, as showed by Pompetchara et al. [28], while the bass had an increased profile of IL6 and IL10. These data, therefore, highlighted the role of these cytokines as predictive biomarkers in disease outcome, that corroborates with Henry et al. [29].

Severe cases also showed increased nitric oxide response and acute inflammatory response, data compatible with the quantification of circulating cytokines in these individuals, in addition to potential responses to opportunistic pathogens such as bacteria and fungi [18]. Macrophages constitute a source of nitric oxide in the body and a high serum nitric oxide is directly related to high macrophages. However, these analyses of opportunistic pathogens not included in this present study and we considerate a limitation this study.

In conclusion, the present work showed that different cellular responses are observed according to the COVID-19 severity in patients from Brazil an epicenter the pandemic in South America. Also, we notice that some cytokines can be used as predictive markers for the disease outcome, possibility implementation of strategies effective by health managers. But it is also important to evaluate the humoral response, since COVID-19 has different outcomes.

## Figures and Tables

**Figure 1 viruses-15-01609-f001:**
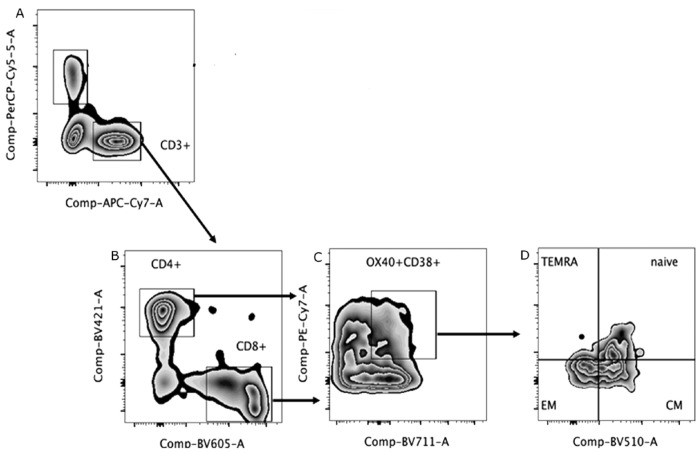
Gate strategy to set T lymphocytes by flow cytometry. (**A**) Gate strategy to set T (CD3+) from peripheral blood mononuclear cells. (**B**) Subsets of TCD4+/TCD8+ according CD3+ and CD4+ or CD8+ expression (**C**) Gating of activated TCD4+/TCD8+ (Ox40+ CD38+); (**D**) Percentage of TEMRA (CD45RA+CCR7-), naïve (CD45RA+CCR7+), EM (CD45RA-CCR7-) and CM (CD45RA-CCR7+) TCD4+/TCD8+.

**Figure 2 viruses-15-01609-f002:**
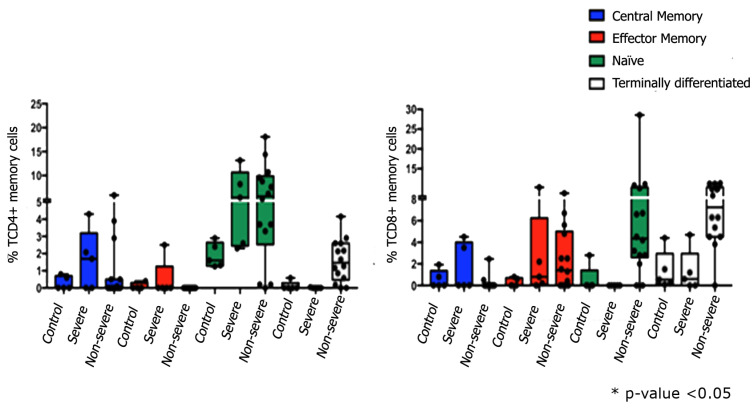
Percentage of activated (CD38^+^OX40^+^). T cells subsets: Central memory T cells TCM CCR7+CD45RA- (blue); Effector memory T cells CCR7-CD45RA- (red); naive T cells CCR7+CD45RA+ (green) and terminally differentiated cells TEMRA CD45RA+CCR7- (white). ANOVA, two-way Kruskal-Wallis test and Dunn’s Multiple Comparison Test was used to perform the comparisons among groups. * *p*-value < 0.05.

**Figure 3 viruses-15-01609-f003:**
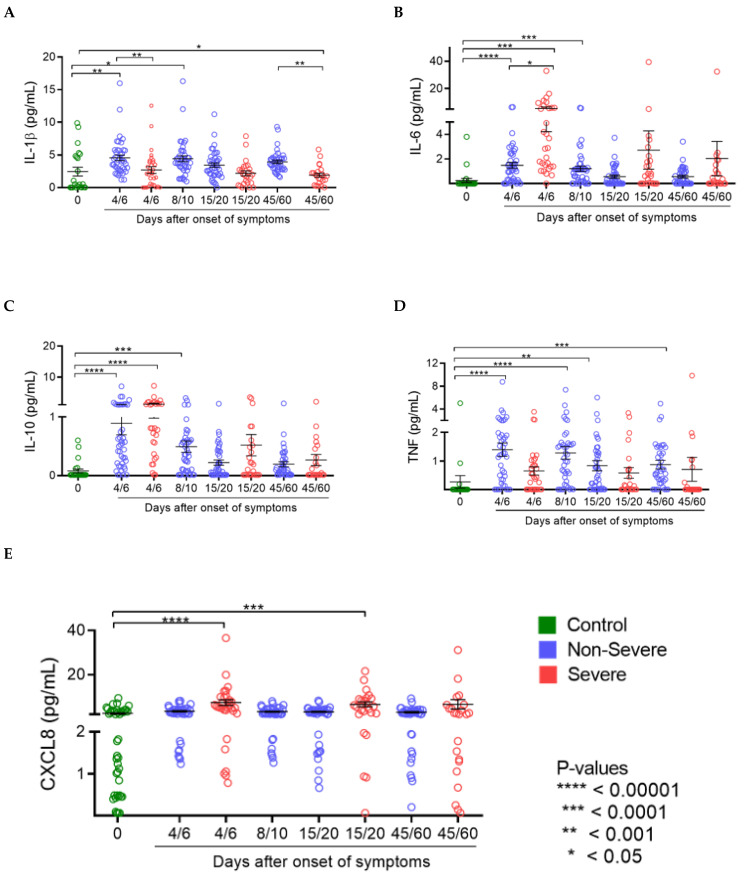
Cytokine levels in participants infected with SARS-CoV-2 and control group non-infected: Comparing control group (Blue) with the Severe (red) and non-Severe (green) groups. (**A**) IL1β; (**B**) IL-6; (**C**) IL-10; (**D**) TNF-α and (**E**) CXCL8 chemokine). ANOVA, two-way Kruskal-Wallis test and Dunn’s Multiple Comparison Test was used to perform the comparisons among groups. *p*-values **** <0.00001, *** <0.0001, ** <0.001 and * <0.05.

**Figure 4 viruses-15-01609-f004:**
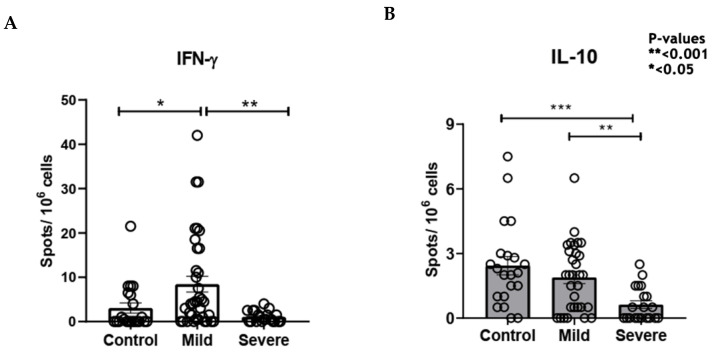
Percentage of cytokine secreting cells from severe and non-severe patients and no infected control group by FluoroSpot data: (**A**) IFN-γ; (**B**) IL10. *p*-values, *** <0.0001, ** <0.001 and * <0.05.

**Table 2 viruses-15-01609-t002:** Laboratory analysis.

Laboratory Analysis	Visit 1		Visit 2		Visit 3		Visit 4	
	Severe	Non Severe	Severe	Non Severe	Severe	Non Severe	Severe	Non Severe
Hemoglobin (g/dL)								
Minimum	8.4	11.3	7.7	12	7.9	11.3	7.4	11.1
Maximum	16.7	16.6	12.4	16.2	15.3	15.9	15.7	16.1
median	13.5	14	10.1	13.9	12.5	13.4	13.1	13.7
Average	13.2	14.3	10.1	14	12.5	13.5	12.8	13.6
Standard deviation	1.9	1.3	3.3	1.1	1.9	1.2	1.7	1.3
Hematocrit (%)								
Minimum	26.6	34.7	25.6	37	26.2	34.6	22.6	34.9
Maximum	49.7	49.9	37.4	48.5	45.8	46.8	45.8	47.2
median	40.2	42.6	31.5	41.7	37.3	40.1	39.4	41.1
Average	40.1	42.8	31.5	41.9	37.8	40.5	38.7	41.1
Standard deviation	5.3	3.7	8.3	3.1	5.2	3.3	5.1	3.2
Global leukocytes (/μL)								
Minimum	3920	2560	6730	3240	5040	3400	5100	3030
Maximum	18500	9920	11920	10740	19040	12910	16860	10720
median	9070	4270	9325	5240	10250	5440	7115	5695
Average	9596.1	4576.1	9325	5739.3	10326	5805.6	7567.7	5758.3
Standard deviation	3198.4	1468.1	3669.9	1915.6	3752.1	1868.1	2553.6	1539.2
Lymphocytes (/μL)								
Minimum	396	726	740.3	907.4	617	1145.5	1180.2	1080
Maximum	3045	2171.5	2264	3494.4	3590	3526.4	3787	3541
median	1093.3	1434.5	1502.2	1785	1663.5	1796	1983	1846.5
Average	1204	1486.2	1502.2	1788.3	1718.6	1864.3	2119	1996.3
Standard deviation	591.4	375.1	1077.4	536.1	665.9	493	659.5	548.2
Platelets (thousand//μL)								
Minimum	142	110	162	135	138	165	44	143
Maximum	603	379	464	494	640	480	413	353
median	270	220	313	254	335	268	268	250.5
Average	304.4	219.2	313	266.8	339.8	282.5	257.5	242.1
Standard deviation	105.2	65.2	213.5	76.7	122.1	66	91.4	51.9
LDH (IU/L)								
Minimum	374.9	138.2	571.8	135.6	243	157.2	238.9	226.6
Maximum	2460.4	632.7	830.5	825	775.9	557.9	684.9	449.2
median	673.2	344.8	701.2	348.3	445.3	321.2	365.7	310.7
Average	777.1	366.7	701.2	358.4	462	335.4	373.7	326.6
Standard deviation	424.7	95.8	182.9	118.5	139.7	71.9	102.3	57.1
Alkaline Phosphatase (IU/L)								
Minimum	113	51.7	134	91	113	88	127	86
Maximum	418	90	245	357	505	361	351	287
median	195	170.5	189.5	182	166	175	170	168
Average	206.3	179	189.5	183.6	195.4	189.2	195.6	175
Standard deviation	75.9	51.7	78.5	54.5	85.3	65	57.3	50.6
TGO/AST (UI/L)								
Minimum	17	12	52	11	10	11	9	11
Maximum	219	116	55	110	65	62	33	63
median	52	24.5	53.5	21	21	20	17.5	18
Average	63.1	28.9	53.5	24.8	27.7	21.8	18.5	20.5
Standard deviation	52.6	16.5	2.1	15.8	16.2	9.3	6.1	9.2
TGP/ALT (UI/L)								
Minimum	11	12	47	9	13	10	9	10
Maximum	691	271	113	355	254	272	97	60
median	72	34.5	80	31	53	25	19	20.5
Average	91.8	39.7	80	41.2	64.4	37.6	24.2	23.8
Standard deviation	121.9	40.2	46.7	52.7	56.2	44.6	17.9	13.1
Ultrasensitive C-reactive protein (mg/L)								
Minimum	2.7	0.4	17.5	0.4	1.5	0.1	0.6	0.1
Maximum	228.4	127.8	142.2	200.9	169.9	18.4	253.2	15.1
median	61	3.8	79.8	1.6	7.6	1.4	3.8	1.3
Average	77.2	12	79.8	14.3	20.8	3.1	16	2.5
Standard deviation	67.2	23.4	88.2	36.9	35.2	4	53.2	2.8
D-dimer (ng/mL)								
Minimum	30	1.7	1325	25	25	25	25	25
Maximum	12968	1244	6360	1981	10636	3671	3040	25000
median	465	31	3842.5	56	420	30	401	30
Average	1451.7	178.3	3842.5	205.2	1441.8	270.3	719.1	744.8
Standard deviation	2734	236.3	3560.3	328	2401	575	903.3	3936
Ferritin (ng/mL)								
Minimum	88.2	24.3	1062.1	4.2	80.2	16.7	25.6	12.3
Maximum	4225	1137	1788	1620.5	1290	910.3	2864.2	365.4
median	1030.5	169.1	1425.1	151.8	598.1	160.2	213	103.6
Average	1300.6	270	1425.1	292.1	593.1	262.9	326.2	130.2
Standard deviation	1115.7	272.2	513.3	336.8	346	247.6	579.2	102.1
Creatinine (mg/dL)								
Minimum	0.5	0.5	1.1	0.5	0.4	0.5	0.5	0.6
Maximum	2.6	1.6	1.2	1.6	6	1.6	2.2	1.7
median	1.1	0.8	1.2	0.8	0.9	0.9	0.8	0.8
Average	1.1	0.8	1.2	0.8	1.2	0.9	0.9	0.9
Standard deviation	0.4	0.2	0.1	0.2	1	0.2	0.3	0.2

## Data Availability

The data that support the findings of this study are available from the corresponding author upon reasonable request.

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
