# Peer review of "Clinical Profile of SARS-CoV-2 Infection: Mechanisms of the Cellular Immune Response and Immunogenetic Markers in Patients from Brazil"

_viruses, 2023, doi:10.3390/v15071609_

Round 1

Reviewer 1 Report

Regarding to the introduction, I suggest to add more information about phenotype and cytokine release in vulnerable group of patients with severe COVID-19. I recommend several articles such as: https://doi.org/10.3390/cancers14184464, https://doi.org/10.3390/jcm12103539.

In regards to CD4 and CD8 T cell phenotype, graphs of Figure 2 should clearly indicate that the percentages of the different phenotypes correspond to activated cells and not to overall CD4 or CD8 T cell populations. In addition, I would include an explanation of the differences observed in the various CD4 and CD8 T cell phenotypes between the three groups.

Figure 3 should be larger and with better quality.

Statistics in Figure 4.

Finally, I would add the last paragraph of the introduction in the discussion.

I recommend checking grammar.

Author Response

Reviewer 1

  1. Regarding to the introduction, I suggest to add more information about phenotype and cytokine release in vulnerable group of patients with severe COVID-19. I recommend several articles such as: https://doi.org/10.3390/cancers14184464,

Authors: Thank you for your suggestion. Some additional information based on their articles were inserted to the text.

  1. In regard to CD4 and CD8 T cell phenotype, graphs of Figure 2 should clearly indicate that the percentages of the different phenotypes correspond to activated cells and not to overall CD4 or CD8 T cell populations. In addition, I would include an explanation of the differences observed in the various CD4 and CD8 T cell phenotypes between the three groups.

Authors: The data shown in the graph are in accordance with the gate strategy in figure 1. In other words, they are data on the percentage of activated T cells, expressing CD38+ and OX40+ and not just percentages of total T CD4+ and T CD8+. To clarify this question, the text and the legend of the figure were modified (lines xxx).

  1. Figure 3 should be larger and with better quality.

Authors: It was modified.

  1. Statistics in Figure 4.

Authors: It was modified.

  1. Finally, I would add the last paragraph of the introduction in the discussion.

Authors: The information about genetics analysis was retired from this manuscript, since it is not concluded.

Reviewer 2 Report

The author showed that different cellular responses are observed according to the COVID-19 severity in patients from Brazil an epicenter the pandemic in South America. This article also confirms that certain cytokines, such as IL-6 and IL-10, may serve as predictive biomarkers for the severity of the disease. The existing data in this manuscript is insufficient to support this conclusion, and there are numerous errors in writing and data presentation. Improvements need to be made in the following aspects:

1. There are some errors in writing or grammar, as follow: i) line 40-41, “Severe COVIS-19 patients, central memory TCD8+ T cells , effector TCD8+ cells …”; ii) line 47, “The entire article uses the consistent spelling of "SARS-CoV-2"; iii) line 269, “The FluoroSpot data demonstrated that 22,8% were found to … ”; ect. Please check the full text for revision and polishing.

2. Table 1 and Table 2 have their positions reversed, and the presentation of the tables is inadequate as they are not in the form of a three-line table. Additionally, the decimal point in the data should be a period instead of a comma.

3. The author's proposition regarding the relationship between the genetic variability of the three major class I histocompatibility complex (MHC) complexes (human leukocyte antigen A [HLA-A], -B, and -C genes) and susceptibility/severity of COVID-19 is crucial, but there is no data presented in the entire article to support this conclusion.

4. The aim of this study was to evaluate the early and late immune responses of patients with SARS-CoV-2 infection during acute infection recovery. However, no consideration was given to whether these cohorts had received the COVID-19 vaccine or the number of doses of the vaccine they had received.

5. The presentation of data in Figure 3 is very non-standard, with the control group presented first and then the experimental group. Additionally, the image quality of the data is poor, with the font size of the x and y-axes being too small. In Figure 4, dot plots should be used instead of bar charts, and the presentation of data should also start with the control group.

6. The median age in this study was 49 years, ranging from 19 to 93 years. However, the article does not differentiate the immune differences after infection between younger and older individuals, which should be analyzed and discussed.

7. Insufficient discussion was provided in the article regarding predictive biomarkers of COVID-19 severity that have been reported in current literature, as well as their similarities and differences with the findings presented in this article.

need polish.

Author Response

Reviewer 2

The author showed that different cellular responses are observed according to the COVID-19 severity in patients from Brazil an epicenter the pandemic in South America. This article also confirms that certain cytokines, such as IL-6 and IL-10, may serve as predictive biomarkers for the severity of the disease. The existing data in this manuscript is insufficient to support this conclusion, and there are numerous errors in writing and data presentation. Improvements need to be made in the following aspects:

  1. There are some errors in writing or grammar, as follow: i) line 40-41, “Severe COVIS-19 patients, central memory TCD8+ T cells , effector TCD8+ cells …”; ii) line 47, “The entire article uses the consistent spelling of "SARS-CoV-2"; iii) line 269, “The FluoroSpot data demonstrated that 22,8% were found to … ”; ect. Please check the full text for revision and polishing.

Authors: It was corrected. However, we didn’t understand the question about “SARS-CoV-2”.

  1. Table 1 and Table 2 have their positions reversed, and the presentation of the tables is inadequate as they are not in the form of a three-line table. Additionally, the decimal point in the data should be a period instead of a comma.

Authors: It was corrected.

  1. The author's proposition regarding the relationship between the genetic variability of the three major class I histocompatibility complex (MHC) complexes (human leukocyte antigen A [HLA-A], -B, and -C genes) and susceptibility/severity of COVID-19 is crucial, but there is no data presented in the entire article to support this conclusion.

Authors: In fact, this work was divided into two stages: Evaluation of immune response mechanisms and analysis of HLA genetic variability. However, studies that address the differences between HLA classes are still in progress, and it will not be possible to include these data in this article. We apologize for the mistaken insertion of this sentence, which was removed from the text because the experiments have not yet been completed.

  1. The aim of this study was to evaluate the early and late immune responses of patients with SARS-CoV-2 infection during acute infection recovery. However, no consideration was given to whether these cohorts had received the COVID-19 vaccine or the number of doses of the vaccine they had received.

Authors: The study was conducted before vaccination. The patients did not receive any doses.

  1. The presentation of data in Figure 3 is very non-standard, with the control group presented first and then the experimental group. Additionally, the image quality of the data is poor, with the font size of the x and y-axes being too small. In Figure 4, dot plots should be used instead of bar charts, and the presentation of data should also start with the control group.

Authors: Both figures were changed.

  1. The median age in this study was 49 years, ranging from 19 to 93 years. However, the article does not differentiate the immune differences after infection between younger and older individuals, which should be analyzed and discussed.

Authors:  The median age in the present study was 49 years, ranging from 19 to 93 years, the genetic data were normalized by age so there was no impact in the genetic expression profile.

  1. Insufficient discussion was provided in the article regarding predictive biomarkers of COVID-19 severity that have been reported in current literature, as well as their similarities and differences with the findings presented in this article.

Authors:  We appreciate your suggestion and added data regarding biomarkers in the discussion (lines 310-313, 325-341, 357-365).

Reviewer 3 Report

The paper presented to me for review addresses the interesting issue of the mechanisms of the cellular immune response and immunogenetic markers of SARS-CoV-2 infection. The paper describes the detailed immunological characteristics of 70 patients with followed up to 60 days. The paper does not raise methodological concerns and is written in good and clear language.

However, before accepting the paper for publication, I would suggest some additions that may improve the quality of the paper:
1. it is worth noting in the introduction that one of the most common symptoms of SARS-CoV-2 infection is headache, which, present during the infection, worsens its course and distant prognosis based on: PMCID: PMC8988454
2. what was the immunological status of the patients analysed? was it one infection or another e.g. second or third illness? were the patients vaccinated?

Author Response

Reviewer 3

The paper presented to me for review addresses the interesting issue of the mechanisms of the cellular immune response and immunogenetic markers of SARS-CoV-2 infection. The paper describes the detailed immunological characteristics of 70 patients with followed up to 60 days. The paper does not raise methodological concerns and is written in good and clear language.

However, before accepting the paper for publication, I would suggest some additions that may improve the quality of the paper:

  1. It is worth noting in the introduction that one of the most common symptoms of SARS-CoV-2 infection is headache, which, present during the infection, worsens its course and distant prognosis based on: PMCID: PMC8988454

Authors: Thank you for your suggestion. It was added to introduction.

  1. What was the immunological status of the patients analyzed? was it one infection or another e.g. second or third illness? were the patients vaccinated?

Authors:  Patients were not vaccinated. The ones without COVID-19, had other diseases, with similar symptoms.

Round 2

Reviewer 1 Report

Suggested changes have been added in the manuscript.

English level is acceptable.

Reviewer 2 Report

The responses answered the questions.